# Effect of In Vitro Digestion on the Antioxidant Compounds and Antioxidant Capacity of 12 Plum (*Spondias purpurea* L.) Ecotypes

**DOI:** 10.3390/foods10091995

**Published:** 2021-08-25

**Authors:** Xochitl Cruz Sollano-Mendieta, Ofelia Gabriela Meza-Márquez, Guillermo Osorio-Revilla, Darío Iker Téllez-Medina

**Affiliations:** Departamento de Ingeniería Bioquímica, Escuela Nacional de Ciencias Biológicas-Zacatenco, Instituto Politécnico Nacional, Av. Wilfrido Massieu S/N, Esq. Cda, Miguel Stampa, Col. Unidad Profesional Adolfo López Mateos, Zacatenco, Alcaldía Gustavo A. Madero, Ciudad de México C.P. 07738, Mexico; xochitl-1221@hotmail.com (X.C.S.-M.); gosorior@ipn.mx (G.O.-R.); darioiker@hotmail.com (D.I.T.-M.)

**Keywords:** bioaccessibility, phenolic compounds, carotenoids, ascorbic acid, ABTS, DPPH

## Abstract

*Spondias purpurea* L. plum is a source of antioxidant compounds. Nevertheless, once they are consumed and go through the digestive system, these compounds may undergo changes that modify their bioaccessibility. This study aimed to evaluate the effect of in vitro gastrointestinal digestion on the total content of carotenoids (TCC), ascorbic acid (AA), phenolic compounds (TPC), flavonoids (TFC), anthocyanins (TAC), and antioxidant capacity (ABTS, DPPH) of 12 plum *Spondias purpurea* L. ecotypes. The plum samples were subjected to the InfoGest in vitro digestion model. TCC, AA, TPC, TFC, TAC, ABTS, and DPPH were significantly different (*p* ≤ 0.05) in each in vitro digestion stage. The gastric stage released the highest content of AA (64.04–78.66%) and TAC (128.45–280.50%), whereas the intestinal stage released the highest content of TCC (11.31–34.20%), TPC (68.61–95.36%), and TFC (72.76–95.57%). Carotenoids were not identified in the gastric stage whilst anthocyanins were lost at the end of the intestinal digestion. At the gastric stage, AA presented a positive and high correlation with ABTS (r: 0.83) and DPPH (r: 0.84), while, in the intestinal stage, TPC and TFC presented positive and high correlation with ABTS (r ≥ 0.8) and DPPH (r ≥ 0.8), respectively.

## 1. Introduction

The plum *S. purpurea* L. originates from Central America and it is distributed from Mexico to Peru and Brazil, although nowadays it has been introduced in certain regions of Asia and Africa. In Mexico, it is considered a native fruit and it is commonly known as Mexican plum or jocote (from náhuatl *xocotl* or acid fruit) [1]. The plum *S. purpurea* L. presents a great diversity of ecotypes (about 30) which are located in different regions of Mexico and differ mainly in the harvesting period, which could occur during the rainy or dry seasons [2].

The fruits of *S. purpurea* L. have a high commercial potential thanks to their low production cost because this plant grows spontaneously and adapts to poor soils in which other plants do not grow, besides, it shows resistance to drought [3]. The fruits have good organoleptic and nutritional qualities as they provide high amounts of carbohydrates, vitamins (A, C), and minerals (potassium, calcium) [2]. In addition, *S. purpurea* L. contains antioxidant compounds such as carotenoids, ascorbic acid, polyphenols, flavonoids, and anthocyanins that can reduce the risks of chronic diseases related to oxidative stress such as cardiovascular illness, diabetes, and obesity [4]. To exert these beneficial effects, the bioactive molecules must reach the site of action; hence, unless they produced their effects directly on the digestive tract, these molecules must be released, absorbed, passed into the blood, and distribute in the organism [5].

The beneficial effects of the antioxidant compounds can be evaluated through the analysis of bioaccessibility, i.e., the amount of compounds released from a matrix into the gastrointestinal tract, becoming available for absorption [6]. During the digestion process, antioxidant compounds can be transformed into other compounds with lower bioaccessibility due to the interactions with other macronutrients such as fiber, proteins, and polysaccharides. Moreover, antioxidant compounds may not be released from the food matrix, which subsequently diminishes their beneficial effects on human health since bioactive compounds produce beneficial effects only when they are available for absorption once the whole digestive process has occurred [7].

The models used to evaluate the bioaccessibility of bioactive compounds can be in vitro (simulated gastrointestinal digestion, artificial membranes, Caco-2 cells culture), ex vivo (laboratory assessment of gastrointestinal organs), in situ (animal intestinal perfusion), in vivo (tests in animals and humans). From these models, in vitro digestion procedures that simulate gastric and small intestinal digestion have been successfully applied to a variety of food matrices to analyze different antioxidant compounds such as carotenoids, phenolic compounds, vitamins (C and E) [8]. The in vitro digestion procedures have the advantage of being fast, low cost, safe, less laborious, and without ethical restrictions that apply to in vivo models. Furthermore, in vitro digestion models allow for higher control of the experimental variables compared to in vivo models. Thus that in vitro simulation is considered more reproducible [9].

Several investigations have analyzed the bioaccessibility of antioxidant compounds in fruits and vegetables through in vitro digestion models [10,11,12]. However, no studies have been found that analyzed antioxidant compounds and antioxidant capacity of the 12 ecotypes of *Spondias purpurea* L. before and after in vitro digestion. Investigations that have been carried out on this fruit are focused on characterization (physical, chemical, and morphological), distribution, agronomical handling, post-harvesting, and functional compounds [2,13,14,15,16]. Therefore, the present study aims to evaluate the influence of in vitro gastrointestinal digestion on the total content of carotenoids, ascorbic acid, phenolic compounds, flavonoids, anthocyanins, and antioxidant capacity (ABTS, DPPH) of the 12 plum (*Spondias purpurea* L.) ecotypes.

## 2. Materials and Methods

### 2.1. Chemicals and Reagents

The following reagents were acquired from Sigma Aldrich Chemical Co. (St. Louis, MO, USA): Folin–Ciocalteu reagent, 1,1-Diphenyl-2-picrylhydrazyl (DPPH), 2,2′-Azino-bis (3-ethylbenzthiazoline-6-sulfonic acid) (ABTS), 6-Hydroxy-2,5,7,8-tetramethylchroman-2-carboxylic acid (Trolox), 2,6-Dichloroindophenol, L-ascorbic acid, gallic acid, quercetin, α-amylase (EC 3.2.1.1), pepsin from porcine gastric mucosa (EC 3.4.23.1), pancreatin from porcine pancreas (EC 232.468.9), and bile extract porcine (EC 232.369.0).

### 2.2. Plum Ecotypes

The 12 ecotypes of Mexican plum were collected in Mexico in the state of Guerrero (17°36′47″ N, 99°57′00″ O) during the months of April and May from the 2020 harvest. One kilogram of fruit at physiological maturity (ready for consumption) of each ecotype was collected, with no evident mechanical damage, pathogens, or damage caused by insects. The ecotypes studied here were chosen from the state of Guerrero because it was reported that they have the highest antioxidant content [2]. Table 1 summarizes the common name and location in which the harvest of the analyzed ecotypes took place.

### 2.3. Sample Preparation

The fruits were washed with a 1% (*v*/*v*) sodium hypochlorite solution, rinsed 3 times with distilled water, and left for drying at room temperature (20 ± 2 °C, 50–60% relative humidity). Afterward, the pulp and epicarp of each fruit was removed (eliminating the seed), then, the samples were stored in suitably labeled, vacuum-sealed polyethylene bags (Selovac, model 200B, São Paulo, Brazil) and kept in an ultra-freezer (Thermo Scientific™ TSX series, Waltham, MA, USA) at −80 °C until analysis. This procedure was carried out on the same day the fruit was collected. The determinations of antioxidant compounds and antioxidant capacity were performed in frozen fruit.

### 2.4. Analytical Methods

#### 2.4.1. Determination of Carotenoids

Extraction and total carotenoid content (TCC) were performed using the Solórzano-Morán et al. [2] methodology. The absorbance was measured at 452 nm using a spectrophotometer (Jenway spectrophotometer, model 7305, Staffordshire, UK). The total carotenoids content was calculated using the absorption coefficient of β-carotene in hexane (2580). The results were expressed as mg of β-carotene per 100 g fresh weight (mg β-carotene/100 g).

#### 2.4.2. Determination of Ascorbic Acid

Extraction and quantification of ascorbic acid (AA) were performed using the method 2,6-dichloroindophenol [17]. The calibration curve (0–0.5 mg/mL, *n* = 6 concentrations) was built using L-ascorbic acid as standard. The absorbance was measured at 520 nm against a xylene blank. The results were expressed as mg of ascorbic acid per 100 g fresh weight (mg AA/100 g). Caution was taken to prevent the loss of AA with the use of reduced lighting and temperature of 4 °C.

#### 2.4.3. Extraction and Determination of Phenolic Compounds

Phenolic compounds extraction was performed by following the method proposed by Kim et al. [18] after minor modifications. One sample of 5 g chopped fruit was homogenized with 10 mL of methanol 60% containing 0.5 mL of HCl 0.1% and placed in a water bath for 2 h, at 85 °C for vitamin C elimination. After cooling, the extract was poured into a 100 mL volumetric flask and completed to volume using distilled water. The extracts were filtered under reduced pressure through a paper filter (Whatman No. 1). The extraction was performed in triplicate. The extracts were packed in glass containers and stored at −20 °C until analyses of phenolic compounds, flavonoids, anthocyanins, and antioxidant capacity, no longer than 3 days.

Total phenolics content (TPC) was determined by the Folin–Ciocalteu method described by Singleton et al. [19]. The calibration curve (0–5 mg/mL, *n* = 7 concentrations) was constructed using gallic acid as standard. The absorbance was measured at 760 nm against a methanol blank. The results were expressed as mg of gallic acid equivalent (GAE) per 100 g fresh weight (mg GAE/100 g).

#### 2.4.4. Determination of Flavonoids

Total flavonoid content (TFC) was calculated according to the colorimetric method described by Zhishen et al. [20]. The calibration curve (0–0.5 mg/mL, *n* = 8 concentrations) was elaborated with quercetin as the standard. The absorbance was measured at 510 nm against a methanol blank. The results were expressed as mg quercetin equivalent (QE) per 100 g fresh weight (mg QE/100 g).

#### 2.4.5. Determination of Anthocyanins

Total anthocyanins content (TAC) was determined using the method by pH difference described by Giusti and Wrolstad [21]. The absorbance was measured at 510 and 700 nm. The total anthocyanins content was calculated using the molecular weight (449.2) and the molar extinction coefficient of cyanidin-3-glucoside (26,900 L). The results were expressed as mg cyanidin-3-glucoside equivalents (C3-GE) per 100 g fresh weight (mg C3-GE/100 g).

#### 2.4.6. Antioxidant Capacity

ABTS assay was performed according to the method described by Re et al. [22]. The calibration curve (0–1500 µM, *n* = 6 concentrations) was elaborated using Trolox as standard. The absorbance was measured at 734 nm against a methanol blank. The results were expressed as mmol Trolox equivalent per 100 g fresh weight (mM TE/100 g). DPPH assay was performed by using the methodology of Brand-Williams et al. [23]. The calibration curve (0–1000 µM, *n* = 8 concentrations) was elaborated using Trolox as standard. The absorbance was measured at 515 nm against a methanol blank. The results were expressed as mmol Trolox equivalent per 100 g fresh weight (mM TE/100 g).

#### 2.4.7. In Vitro Digestion

The bioaccessibility of antioxidant compounds was analyzed using an in vitro gastrointestinal digestion procedure, which intends to simulate the physiological process in the digestive tract (mouth, stomach, and small intestinal digestion sequentially). The in vitro digestion model was followed as described by Minekus et al. [9].

The oral stage was simulated by grinding 5 g of sample with 3.5 mL of simulated salivary fluid (pH 7), 0.5 mL of α-amylase solution (75 U/mL), 25 µL of CaCl_2_ (0.3 M), and 975 µL of distilled water. The mixture was placed into 50 mL Falcon conical tubes and they were agitated at 100 rpm at 37 ± 2 °C for 2 min, whereas pH was adjusted at 7 using NaOH (1 M). To simulate gastric digestion, the oral bolus was mixed with 7.5 mL of simulated gastric fluid (pH 2.5), 1.6 mL of pepsin from porcine gastric mucosa (2000 U/mL), 5 µL of CaCl_2_ (0.3 M), and 690 µL of distilled water, while pH was adjusted and kept at 2.5 with HCl (1 M). Afterward, the mix was agitated at 100 rpm at 37 ± 2 °C for 2 h. The small intestine stage was simulated by mixing the gastric chyme with 5.5 mL of simulated intestinal fluid (pH 7), 2.5 mL of pancreatin from porcine pancreas (800 U/mL), 1.25 mL porcine bile extract (10 mM), 20 µL de CaCl_2_ (0.3 M) and 585 µL of distilled water, while pH was adjusted and kept at 7 with NaOH (1 M). Then, the mixture was incubated at 37 ± 2 °C at 100 rpm for 2 h.

At the end of each stage, gastric (120 min) and intestinal (120 min), aliquots were taken and placed in an ice bath for 10 min to stop the enzymatic reaction. Subsequently, the aliquots (from gastric and intestinal stages) were centrifuged at 3000× *g* for 20 min. Finally, the supernatant was used to analyze TCC, AA, TPC, TFC, TAC, ABTS, and DPPH. Every simulation was carried out in triplicate. The in vitro digestion was performed in the absence of light and oxygen, i.e., the falcon tubes used were hermetically closed and covered with aluminum foil. It has been shown that anaerobic in vitro systems can simulate the physiological conditions in the stomach and small intestine with higher efficiency than the aerobic digestions [24].

The effect of in vitro gastrointestinal digestion on the content of TCC, AA, TPC, TFC, TAC, ABTS, and DPPH was evaluated with the bioaccessibility index (*BI*) calculated with Equation (1) [25]:(1)$$BI\;\left(\%\right)=\frac{A}{B}\times 100$$
where *A* is the amount of each compound (TCC, AA, TPC, TFC, TAC) or antioxidant capacity (ABTS, DPPH) quantified in each digestion phase (gastric, intestinal), *B* is the amount of each compound (TCC, AA, TPC, TFC, TAC) or antioxidant capacity (ABTS, DPPH) quantified in the undigested food matrix.

#### 2.4.8. Statistical Analysis

All measurements were conducted in triplicate. The results were analyzed by descriptive statistics (mean and standard deviation), one-way analysis of variance (ANOVA), and comparison of means using the Tukey method, with a significance level of 5%. Principal component analysis (PCA) was performed to infer the relationship among variables. The correlation between antioxidant capacity (ABTS, DPPH) and antioxidant compounds (TCC, AA, TPC, TFC, TAC) before and after the in vitro digestion was determined through Pearson correlation tests. The statistical analysis was performed with the software Minitab version 16.1.0 (State College, PA, USA).

## 3. Results

### 3.1. Total Carotenoids Content before and after In Vitro Digestion

The effect of in vitro gastrointestinal digestion on the carotenoids content is presented in Table 2. Before in vitro digestion, the total carotenoids content expressed as β-carotene ranged from 1.24 ± 0.36 mg β-carotene/100 g (Chicamerito) to 2.91 ± 0.04 mg β-carotene/100 g (Carnuda). The carotenoid content found in the 12 ecotypes of de *S. purpurea* L. coincided with that reported in other studies [2,26]. Studies indicated that the carotenoid content of *S. purpurea* L. was higher than that found in other fruits such as soursop (0.39 mg/100 g), strawberry (0.46 mg/100 g), and blackberry (1.02 mg/100 g) [26]. The epicarp of several ecotypes of *S. purpurea* L. has a wide variety of colors such as red, purple, orange, yellow, and green, and these colors have been attributed mainly to the presence of phenolic compounds and carotenoids [2].

After gastric digestion, carotenoids were not detected for all of the plum ecotypes, which could be attributed to the amount of pectin present in the fruits. Studies have shown the negative effect of pectin on the release and bioaccessibility of carotenoids [27,28] by intervening in the digestion of lipids, which help to release carotenoids from emulsified lipid droplets in the gastrointestinal content and for micelles formation, in which carotenoids must be incorporated before absorption. Only micellized carotenoids may be absorbed and subsequently exert protective effects on human health [27]. Cervantes et al. [28] reported that carotenoid micellization decreases in fruits with a high pectin content. On the contrary, this process is favored in fruits with low pectin content. Therefore, the amount of pectin present in the fruits seems to be related to the carotenoid release since pectin could interact with carotenoids and thus affect its release to the digestive medium due to its intragastric gelation properties [29].

Unlike the gastric stage, after intestinal digestion, carotenoids were detected (0.19 ± 0.16–0.68 ± 0.03 mg β-carotene/100 g) in all plum ecotypes. However, their content was significantly (*p* ≤ 0.05) lower than that quantified by chemical extraction, which could be due to several factors interfering in the digestive process, such as enzymes, pH, and temperature. Enzymes (α-amylase, pepsin, pancreatin), temperature (37 °C), and changes in pH (2.5 or 7) during in vitro digestion can influence the release of antioxidant compounds in different ways, and this will depend on the food matrix and the interaction with other compounds such as proteins, carbohydrates, lipids, fiber, or minerals [6].

As previously mentioned, carotenoids were quantified after intestinal digestion in comparison with gastric digestion. These results suggest that carotenoids could be retained by the pectin gel during gastric incubation, and they were released when the medium conditions changed, as is the case of the change from the gastric (pH 2.5) to the intestinal medium (pH 7). It could also be attributed to the addition of pancreatin (800 U/mL) during the intestinal stage, which has been reported to potentialize carotenoids release. Garret et al. [30] reported that the absence of pancreatin during the intestinal stage decreases carotenoids release to 50%. On the contrary, high pancreatin concentrations favor carotenoids release.

Ecotype Atoyaxi presented the lowest BI (11.31%), and Guingure amarilla presented the highest BI (34.20%). This would indicate that carotenoids belonging to the ecotype Guingure amarilla were more bioaccessible or suitable to be absorbed in the small intestine and would exert beneficial effects on health, though more studies are required. The bioaccessibility of carotenoids depends on several factors such as the fruit maturity state [6]. In addition, cooking influences carotenoids’ bioaccessibility. Studies have shown that the bioaccessibility of carotenoids from cooked fruits is higher than that from raw fruits since thermal processing enhances cell-wall breakdown and carotenoid release [6,31]. Freezing the fruit before the analysis also favors the release and bioaccessibility of carotenoids [31]. Another factor that influences carotenoid release is the presence of lipids since they create a lipophilic environment that enhance carotenoid bioaccessibility. In addition, lipid hydrolysis modifies physicochemical characteristics of micelles, which can increase carotenoid uptake. Generally, *S. purpurea* L. is not consumed along with lipids; nevertheless, this fruit is usually consumed after a meal with lipids, which may have a positive effect on carotenoids bioaccessibility. Studies indicate that carotenoids’ bioaccessibility in mixed diets is 28%, whilst, in lipid diets, carotenoids bioaccessibility is 53% [31].

### 3.2. Content of Ascorbic Acid before and after In Vitro Digestion

Total ascorbic acid content obtained through chemical extraction ranged from 23.81 ± 2.01 mg AA/100 g to 29.36 ± 4.01 mg AA/100 g (Table 3). Among the studied ecotypes, the lowest ascorbic acid content was found in Costeña (23.81 ± 2.01 mg AA/100 g), Conservera (25.07 ± 5.01 mg AA/100 g), and Chabacana (26.31 ± 2.01 mg AA/100 g), whilst Guingure amarilla presented the highest content of ascorbic acid (29.36 ± 4.01 mg AA/100 g). These results of ascorbic acid were comparable with other studies [14,32]. *S. purpurea* L. is a good source of ascorbic acid since its content is higher than that found in other fruits such as papaya (8.6 mg AA/100 g), nanche (11.8 mg AA/100 g), and pineapple (13.0 mg AA/100 g) [32].

In each plum ecotype, the content of ascorbic acid obtained through chemical extraction was significantly (*p* < 0.05) higher in comparison to that obtained after gastric digestion (18.15 ± 1.17–19.99 ± 1.12 mg AA/100 g) and intestinal digestion (6.04 ± 1.05–8.41 ± 1.08 mg AA/100 g), which could be attributed to several factors involved in the digestive process such as enzymes, pH, and temperature. Research works have reported losses of ascorbic acid after digestion with pepsin [33]. However, the BI of ascorbic acid after gastric digestion was higher (64.04–78.66%) than that obtained after intestinal digestion (22.60–29.06%). This is probably due to the acidity (pH 2.5) of the gastric medium, which protected the ascorbic acid against enzymatic oxidation. Studies have reported that ascorbic acid was more stable in acidic conditions than in alkaline [6]. During the gastric stage, the hydrochloric acid lowered the pH to 2.5, which might help to increase the ascorbic acid stability. The loss of ascorbic acid during intestinal digestion could be due to pH (7) changes and the presence of oxygen after the enzymatic digestion process contributing to the ascorbic acid degradation. The ascorbic acid sensitivity to the small intestine conditions was previously reported [33].

### 3.3. Total Phenolic Compounds Content before and after In Vitro Digestion

Total phenolic content obtained by chemical extraction ranged from 326.09 ± 2.94 GAE/100 g (Chabacana) to 620.18 ± 2.14 mg GAE/100 g (Morada) (Table 4). The total phenolic content of the *S. purpurea* L. 12 ecotypes matched data reported in other works [26]. Studies indicated that phenols present in *S. purpurea* L. have a natural antioxidant function and its consumption brings benefits against chronic and degenerative illnesses. Furthermore, *S. purpurea* L. is a good source of polyphenols since it is higher than the one found in other fruits such as papaya (54 mg GAE/100 g), banana (24–72 mg GAE/100 g), pineapple (35–52 mg GAE/100 g), soursop (154–284 mg GAE/100 g), and yellow plum *Spondias mombi* (27.0 mg GAE/100 g) [2,16].

*S. purpurea* L. shows different colors depending on the ecotype. It was reported that red and purple ecotypes have the highest phenolic content in comparison to those that are yellow or orange [2,3], the above agrees with the results found in the present work since the ecotypes with red epicarp (Atoyaxi, Carnuda, Chicamerito, Costeña, Cuernavaqueña, Guingure roja, Morada) and red-purple epicarp (Conservera) reported the highest content of phenolic compounds.

After gastric digestion, the phenolic content decreased significantly (*p* < 0.05) in comparison to the initial phenolic content (chemical extraction) for all the plum ecotypes. This could be due to the contribution of enzymes, temperature, and pH during gastric digestion, as well as to the use of methanol in the chemical extraction to release phenolic compounds more efficiently [34]. The decrease in phenolic content after gastric digestion could also be attributed to the pectin present in the fruit. Studies have suggested that pectin could interact with phenolic compounds and affect its release to the digestive medium due to its intragastric gelation properties [29]. *S. purpurea* L. has a pectin content of 4%. This quantity is enough to promote gel formation in acidic conditions, which could explain the decrease in phenolic compounds content in the gastric stage. This same behavior was observed in fruits with high amounts of pectin, which presented a lower phenolic compound release after gastric digestion [35].

After intestinal digestion, the number of phenolic compounds increased significantly (*p* < 0.05), i.e., from the gastric (213.44 ± 3.22–282.92 ± 2.71 mg GAE/100 g) to the intestinal stage (241.57 ± 3.39–511.75 ± 2.95 mg GAE/100 g) for all the plum ecotypes. Nevertheless, this amount was significantly (*p* < 0.05) lower than that quantified by chemical extraction, which could indicate that a certain amount of phenolic compounds was released to transform into different structures with other chemical properties and biological activity. For example, phenolic compounds could be bound to proteins through several mechanisms such as hydrogen, covalent, or hydrophobic bonds [29]. This behavior agrees with Bouayed et al. [35], who reported a smaller release of phenolic compounds in the gastric stage and, afterward, an increment in the intestinal one.

The BI of phenolic compounds after gastric digestion ranged from 37.70 to 77.57%, whereas the BI ranged from 68.62 to 95.36% after intestinal digestion. Ecotype Atoyaxi presented the lowest BI (68.61%) and Costeña the highest BI (95.36%). This would indicate that the phenolic compounds from the latter would be more bioavailable to be absorbed in the small intestine and would exert greater beneficial health effects in comparison to other ecotypes. Differences in the BI of phenolic compounds among the different *S. purpurea* L. ecotypes were probably due to the binding of these compounds, during digestion, to other molecules such as proteins, minerals, and polysaccharides. Moreover, the food matrix, gastrointestinal digestion conditions (pH, enzymes, temperature, among others), and the methodology used to quantify phenolic compounds also influenced the phenolic compounds bioaccessibility. Another probable cause is the affinity of the Folin–Ciocalteu reagent since it can react with non-phenolic compounds, such as reducing sugars, vitamins, amino acids, and proteins, and this method could overestimate the phenolic compounds [18].

### 3.4. Total Flavonoid Content before and after In Vitro Digestion

Total flavonoid content obtained through chemical extraction ranged from 195.95 ± 0.96 mg QE/100 g (Carnuda) to 368.78 ± 1.70 mg QE/100 g (Costeña) (Table 5). Flavonoid content agrees with other works, which reported that it was higher than that found in other tropical fruits such as grape (55.9 mg QE/100 g), papaya (63.2 mg QE/100 g), acai (70.1 mg QE/100 g), strawberry (21.8 mg QE/100 g), and plum *Spondias mombi* (8.7 mg QE/100 g) [11,12]. Studies indicated that *S. purpurea* L. flavonoids have antioxidant activity and a positive effect to prevent cardiovascular disorders and illnesses caused by free radicals [36].

Moo-Huchin et al. [36] reported high phenolic content and low quantities of flavonoids, carotenoids, and ascorbic acid in *S. purpurea* L. harvested in Yucatán, México. These results agreed with the present study since the phenolic content (326.09 ± 2.94–620.18 ± 2.14 mg GAE/100 g) was greater than the flavonoid content (195.95 ± 0.96–368.78 ± 1.70 mg QE/100 g), carotenoids (1.24 ± 0.36–2.91 ± 0.04 mg β-carotene/100 g), and ascorbic acid (23.81 ± 2.01–29.36 ± 4.01 mg AA/100 g) in the 12 ecotypes.

After the simulated gastrointestinal digestion, the compound release was similar to the one found for phenolic compounds, i.e., gradual release from the gastric to the intestinal stage. However, flavonoid content after gastric and intestinal stages was significantly (*p* < 0.05) lower than that determined using chemical extraction in all plum ecotypes. This behavior can be due to pH changes, interaction with other released compounds such as iron, dietary fiber, proteins, and digestive enzymes that enable flavonoid release from the food matrix. Studies indicated that flavonoids associated with high molecular weight compounds such as proteins and carbohydrates can be released in the digestive system through enzymatic action [12]. The BI of flavonoids after gastric digestion ranged from 31.86 to 65.10%, whilst BI after intestinal digestion ranged between 72.76 and 95.57%. Flavonoid bioaccessibility varied accordingly to the gastric and intestinal conditions (pH, enzymes) and the capability to binding proteins or fiber through hydrogen or covalent bonds or hydrophobic interactions [12].

### 3.5. Total Anthocyanins Content before and after In Vitro Digestion

Total anthocyanins content by chemical extraction ranged from 2.00 ± 0.29 mg C3-GE/100 g (Silvestre) to 14.92 ± 1.19 mg C3-GE/100 g (Atoyaxi) (Table 6). These results agreed with other studies [32]. In ecotypes with yellow epicarp (Chabacana, Guingure amarilla, and Amarilla) no anthocyanins were detected through chemical extraction, this was because the color of *S. purpurea* L. was mainly attributed to phenols and carotenoids (pigments responsible for yellow-red color), whilst anthocyanins were pigments with color spanning from red to blue. There are few studies on anthocyanins content in plum *S. purpurea* L. [32].

The BI of anthocyanins after gastric digestion was elevated (128.45–280.50%). Previous studies reported high bioaccessibility of anthocyanins (≥100%) after gastric digestion, and this was related to their chemical characteristics such as solubility, hydrophobicity, molecular weight, and isomer configuration that make anthocyanins molecules resistant to acidic media [35]. In addition, the high bioaccessibility of anthocyanins was related to the protection effect by other specific compounds present in the food matrix, for instance, soluble and insoluble fiber, that interact with anthocyanins by increasing their stability during the digestion process [12]. Studies indicate that, in acidic media, anthocyanins may show higher color intensity due to interaction with colorless flavonols through co-pigmentation phenomena [37].

On the other hand, after intestinal digestion, anthocyanins were not detected for all plum ecotypes. This is due to the intense degradation anthocyanins undergo after the duodenal stage, as reported by other studies, which found a significant decrease of anthocyanins content after the duodenal stage [12,35]. The above can be attributed to the fact that anthocyanins are unstable at higher pH, for instance, the pH in the intestinal medium (>6). Furthermore, the high instability of anthocyanins at the pH of the intestine medium is attributable to the formation of the colorless pseudo base chalcone, which destroys the anthocyanin chromophore.

### 3.6. Antioxidant Capacity before and after In Vitro Digestion

Antioxidant capacity by chemical extraction using ABTS was higher (5.18 ± 0.13–8.23 ± 0.06 mM TE/100 g) in comparison with the one obtained through DPPH (1.71 ± 0.06–1.89 ± 0.02 mM TE/100 g) for all the plum ecotypes (Table 7 and Table 8). This might be observed because ABTS measured hydrophilic and lipophilic antioxidants, whilst DPPH only measured hydrophobic compounds.

After in vitro digestion, antioxidant capacity with both methods (ABTS, DPPH) increased significantly (*p* < 0.05) during the intestinal stage in comparison to the findings during the gastric stage for all plum ecotypes. This behavior could be due to pH variation during in vitro digestion. Previous studies suggest that the transition from the acidic medium to alkaline increases the phenolic compounds and flavonoids release, which contributes to the increase of antioxidant activity. This occurs because of deprotonation of hydroxyl groups present in aromatic rings [11,12]. Another aspect that influences antioxidant capacity is the interaction of phenolic compounds with other compounds released during digestion, such as minerals or dietary fiber, that affect solubility and phenols availability. Furthermore, non-antioxidant food components such as amino acids, sugars, and uronic acids can be released after in vitro gastrointestinal digestion and show positive interference in Trolox equivalent antioxidant capacity assays [6]. These results indicate that the 12 ecotypes of *S. purpurea* L. are a good source of antioxidants before and after in vitro digestion. 

### 3.7. Correlation between Antioxidant Compounds and Antioxidant Capacity before and after In Vitro Digestion

Pearson correlation coefficients among the antioxidant capacity (ABTS y DPPH) and antioxidant compounds before and after in vitro gastrointestinal digestion are presented in Table 9. Before in vitro digestion, ABTS and DPPH presented high and positive significant correlation (*p* < 0.01) with phenolic compounds (r = 0.94 and 0.83, respectively), flavonoids (r = 0.91 y 0.82, respectively), and anthocyanins (r = 0.82 y 0.73, respectively). Whereas carotenoids (r = 0.44–0.62, *p* < 0.01) and ascorbic acid (r = 0.65–0.43, *p* < 0.01) showed moderated correlation with both antioxidant methods. Before in vitro digestion, the contents of phenolic compounds, flavonoids, and anthocyanins were high and could be the most important contributors to the antioxidant capacity of the 12 ecotypes of *S. purpurea* L. After the gastric stage, only ascorbic acid presented a moderate correlation with ABTS (r = 0.83, *p* < 0.01) and DPPH (r = 0.84, *p* < 0.01). No correlation was obtained to any of the rest of the antioxidant compounds. These results indicate that, during the gastric stage, ascorbic acid contributed to the antioxidant capacity. However, this moderate correlation decreased after the intestinal stage, as previously discussed.

After the intestinal stage, total phenolics and flavonoids presented high correlation with ABTS (r > 0.8, *p* < 0.01) and DPPH (r > 0.8, *p* < 0.01). Ascorbic acid had low correlation with ABTS (r = 0.44, *p* < 0.01) and DPPH (r = 0.29, *p* < 0.01). No correlation was found between the contents of other antioxidant compounds and antioxidant capacity. The above suggests that in the intestinal stage, the highest release of phenolic and flavonoids occurred and, therefore, the highest correlation with the antioxidant capacity, independently of the method used.

The results indicate that, before and after the intestinal digestion, antioxidant capacity by the two assays in the 12 ecotypes of *S. purpurea* L. can be mainly attributed to phenolics and flavonoids. Research works indicate that the antioxidant capacity of flavonoids could be due to the presence of double bonds in the C ring, which potentializes the nucleophilic power, while the antioxidant capacity from the phenolic acids depends on the number of hydroxyl groups within the molecule [26].

Different studies have reported relationships among phenolic compounds and the antioxidant capacity of fruits. While some authors found a high correlation [3,26,32], others do not report a relation between the phenolic content and the antioxidant capacity [2]. The above suggests that each phenolic compound can contribute in a different way and proportion and it is necessary to consider that such correlation depends on many factors such as phenolic content, antioxidant quality, interaction with other components, and applied methodology. Moo-Huchin et al. [36] indicate that high or null correlations between the phenolic compounds and antioxidant capacity could be explained by the presence of reducing agents such as ascorbic acid, minerals, and carotenoids in fruits.

## 4. Conclusions

This research has reported for the first time the effect of in vitro gastrointestinal digestion on antioxidant compounds (carotenoids, ascorbic acid, polyphenols, flavonoids, anthocyanins) and the antioxidant capacity of 12 plum (*S. purpurea* L.) ecotypes. Each antioxidant compound presented a characteristic behavior during the in vitro digestion process. The ascorbic acid and anthocyanins were released mainly during gastric digestion, whilst carotenoids, polyphenols, and flavonoids were released mainly during intestinal digestion. The most important factor that affects the release of antioxidant compounds seems to be the gastrointestinal tract medium (pH, enzymes, temperature). Results from the present work contribute to the knowledge of antioxidant compounds of 12 plum (*S. purpurea* L.) ecotypes. Although these in vitro results cannot be extrapolated to in vivo conditions, these data provide evidence that the different ecotypes of *S. purpurea* L. studied constitute rich sources of antioxidant compounds. The findings from this study could help agronomists who are looking to select or produce *S. purpurea* L. ecotypes with the highest amount of antioxidant compounds with beneficial health effects.

It is recommended to perform more studies to know the effect of simulated in vitro digestion over the individual profile of carotenoids and phenolic compounds present in the 12 ecotypes *S. purpurea* L. Likewise, the individual phenolic compounds of the 12 ecotypes grown in Mexico must be identified by using some methods such as High-Performance Liquid Chromatography (HPLC). It is advised to conduct in vivo studies to confirm whether the simulated gastrointestinal digestion model presented in this study generates or not similar conclusions on the bioaccessibility of antioxidant compounds of the 12 ecotypes *S. purpurea* L. studied here.

## Figures and Tables

**Table 1 foods-10-01995-t001:** Sampling location and epicarp color of the 12 analyzed ecotypes of *S. purpurea* L.

Ecotype	Sampling Location in Mexico	Epicarp Color
Atoyaxi	Jojutla, Guerrero	Red
Carnuda	Chilpancingo, Guerrero	Red
Chabacana	Sinahua, Guerrero	Yellow-orange
Chicamerito	Chicamerito, Guerrero	Red
Costeña	Tierra colorada, Guerrero	Red
Cuernavaqueña	Chilpancingo, Guerrero	Red
Guingure amarilla	Cd. Altamirano, Guerrero	Yellow
Guingure roja	Cd. Altamirano, Guerrero	Red
Morada	Chilpancingo, Guerrero	Red
Conservera	Chilpancingo, Guerrero	Red-purple
Amarilla	Chilpancingo, Guerrero	Yellow
Silvestre	Chilpancingo, Guerrero	Orange

**Table 2 foods-10-01995-t002:** Total carotenoid content and bioaccessibility index of 12 ecotypes of *S. purpurea* L. before and after in vitro digestion.

Ecotype	Chemical Extraction (mg β-Carotene/100 g)	Gastric Phase (mg β-Carotene/100 g)	BI (%)	Intestinal Phase (mg β-Carotene/100 g)	BI (%)
Atoyaxi	1.68 ± 0.65 ^bc, A^	ND	0.00	0.19 ± 0.16 ^d, B^	11.31
Carnuda	2.91 ± 0.04 ^a, A^	ND	0.00	0.46 ± 0.05 ^abcd, B^	15.81
Chabacana	2.21 ± 0.32 ^abc, A^	ND	0.00	0.48 ± 0.02 ^abc, B^	21.72
Chicamerito	1.24 ± 0.36 ^c, A^	ND	0.00	0.32 ± 0.22 ^bcd, B^	25.81
Costeña	1.82 ± 0.16 ^abc, A^	ND	0.00	0.35 ± 0.08 ^bcd, B^	19.23
Cuernavaqueña	2.57 ± 0.70 ^ab, A^	ND	0.00	0.68 ± 0.03 ^a, B^	26.46
Guingure amarilla	1.93 ± 0.12 ^abc, A^	ND	0.00	0.66 ± 0.12 ^a, B^	34.20
Guingure roja	2.55 ± 0.41 ^ab, A^	ND	0.00	0.54 ± 0.08 ^ab, B^	21.18
Morada	1.37 ± 0.44 ^c, A^	ND	0.00	0.24 ± 0.09 ^cd, B^	17.52
Conservera	2.78 ± 0.37 ^ab, A^	ND	0.00	0.53 ± 0.09 ^ab, B^	19.06
Amarilla	2.55 ± 0.17 ^ab, A^	ND	0.00	0.64 ± 0.09 ^a, B^	25.10
Silvestre	2.14 ± 0.12 ^abc, A^	ND	0.00	0.58 ± 0.08 ^ab, B^	27.10

ND: Not detected. Values are expressed as the mean ± standard deviation. Different lowercase letters per column indicate a significant statistical difference (*p* ≤ 0.05). Different capital letters per row a indicate significant statistical difference (*p* ≤ 0.05). BI: bioaccessibility index.

**Table 3 foods-10-01995-t003:** Total ascorbic acid content and bioaccessibility index of 12 ecotypes of *S. purpurea* L. before and after in vitro digestion.

Ecotype	Chemical Extraction (mg AA/100 g)	Gastric Phase (mg AA/100 g)	BI (%)	Intestinal Phase (mg AA/100 g)	BI (%)
Atoyaxi	28.21 ± 1.97 ^a, A^	18.88 ± 1.32 ^a, B^	66.93	7.91 ± 1.35 ^a, C^	28.04
Carnuda	26.52 ± 0.84 ^a, A^	19.99 ± 1.12 ^a, B^	75.38	6.73 ± 0.83 ^a, C^	25.38
Chabacana	26.31 ± 2.01 ^a, A^	19.16 ± 0.98 ^a, B^	72.82	7.51 ± 1.02 ^a, C^	28.54
Chicamerito	27.60 ± 0.66 ^a, A^	18.15 ± 1.17 ^a, B^	65.76	7.20 ± 1.04 ^a, C^	26.09
Costeña	23.81 ± 2.01 ^a, A^	18.73 ± 0.57 ^a, B^	78.66	6.49 ± 1.07 ^a, C^	27.26
Cuernavaqueña	26.46 ± 3.01 ^a, A^	18.63 ± 0.68 ^a, B^	70.41	7.22 ± 1.12 ^a, C^	27.29
Guingure amarilla	29.36 ± 4.01 ^a, A^	19.73 ± 0.46 ^a, B^	67.20	8.41 ± 1.08 ^a, C^	28.64
Guingure roja	28.39 ± 4.56 ^a, A^	18.18 ± 0.71 ^a, B^	64.04	6.77 ± 0.79 ^a, C^	23.85
Morada	27.13 ± 4.07 ^a, A^	18.31 ± 1.01 ^a, B^	67.49	6.53 ± 1.08 ^a, C^	24.07
Conservera	25.07 ± 5.01 ^a, A^	18.86 ± 0.89 ^a, B^	75.23	7.12 ± 1.07 ^a, C^	28.40
Amarilla	26.72 ± 4.88 ^a, A^	19.86 ± 1.02 ^a, B^	74.33	6.04 ± 1.05 ^a, C^	22.60
Silvestre	28.53 ± 4.71 ^a, A^	18.99 ± 1.01 ^a, B^	66.56	8.29 ± 1.25 ^a, C^	29.06

Values are expressed as the mean ± standard deviation. Different lowercase letters per column indicate a significant statistical difference (*p* ≤ 0.05). Different capital letters per row indicate a significant statistical difference (*p* ≤ 0.05). BI: bioaccessibility index.

**Table 4 foods-10-01995-t004:** Total phenolic content and bioaccessibility index of 12 ecotypes of *S. purpurea* L. before and after in vitro digestion.

Ecotype	Chemical Extraction (mg GAE/100 g)	Gastric Phase (mg GAE/100 g)	BI (%)	Intestinal Phase (mg GAE/100 g)	BI (%)
Atoyaxi	431.57 ± 2.71 ^d, A^	233.55 ± 3.02 ^de, C^	54.12	296.09 ± 3.21 ^fg, B^	68.61
Carnuda	464.44 ± 3.32 ^c A^	261.01 ± 3.42 ^c, C^	56.20	403.40 ± 3.24 ^c, B^	86.86
Chabacana	326.09 ± 2.94 ^i, A^	215.51 ± 2.65 ^f, C^	66.09	304.07 ± 4.23 ^f, B^	93.25
Chicamerito	596.12 ± 2.41 ^b, A^	236.93 ± 2.41 ^de, C^	39.75	465.97 ± 2.18 ^i, B^	78.17
Costeña	461.11 ± 3.18 ^c, A^	282.92 ± 2.71 ^a, C^	61.36	439.73 ± 1.23 ^b, B^	95.36
Cuernavaqueña	344.82 ± 3.27 ^g, A^	213.44 ± 3.22 ^f, C^	61.90	276.69 ± 1.14 ^h, B^	80.24
Guingure amarilla	364.98 ± 1.71 ^f, A^	230.20 ± 3.91 ^e, C^	63.07	281.01 ± 2.35 ^h, B^	76.99
Guingure roja	379.58 ± 1.63 ^e, A^	272.48 ± 3.62 ^b, C^	71.78	354.98 ± 3.74 ^d, B^	93.52
Morada	620.18 ± 2.14 ^a, A^	233.81 ± 3.38 ^de, C^	37.70	511.75 ± 2.95 ^a, B^	82.52
Conservera	332.06 ± 3.24 ^hi, A^	240.25 ± 2.02 ^d, C^	72.35	291.08 ± 2.23 ^g, B^	87.66
Amarilla	334.69 ± 2.47 ^h, A^	233.55 ± 3.71 ^de, C^	69.78	241.57 ± 3.39 ^j, B^	72.18
Silvestre	339.31 ± 3.07 ^gh, A^	263.20 ± 3.53 ^bc, C^	77.57	314.28 ± 2.49 ^e, B^	92.62

Values are expressed as the mean ± standard deviation. Different lowercase letters per column indicate significant statistical difference (*p* ≤ 0.05). Different capital letters per row indicate significant statistical difference (*p* ≤ 0.05). BI: bioaccessibility index.

**Table 5 foods-10-01995-t005:** Total flavonoid content and bioaccessibility index of 12 ecotypes of *S. purpurea* L. before and after in vitro digestion.

Ecotype	Chemical Extraction (mg QE/100 g)	Gastric Phase (mg QE/100 g)	BI (%)	Intestinal Phase (mg QE/100 g)	BI (%)
Atoyaxi	214.70 ± 1.76 ^h, A^	121.72 ± 1.15 ^fg, C^	56.69	169.30 ± 1.55 ^g, B^	78.85
Carnuda	195.95 ± 0.96 ^i, A^	124.86 ± 1.40 ^ef, C^	63.72	144.37 ± 1.80 ^h, B^	73.68
Chabacana	245.84 ± 4.49 ^g, A^	145.78 ± 3.68 ^b, C^	59.30	205.15 ± 1.62 ^e, B^	83.45
Chicamerito	346.18 ± 1.50 ^b, A^	172.74 ± 3.41 ^a, C^	49.90	294.85 ± 1.21 ^a, B^	85.17
Costeña	368.78 ± 1.70 ^a, A^	117.50 ± 1.25 ^gh, C^	31.86	281.64 ± 4.78 ^b, B^	76.37
Cuernavaqueña	321.37 ± 3.25 ^c, A^	121.14 ± 3.82 ^fg, C^	37.69	263.24 ± 3.84 ^c, B^	81.91
Guingure amarilla	274.51 ± 3.36 ^e, A^	103.44 ± 1.44 ^i, C^	37.68	213.07 ± 4.05 ^e, B^	77.62
Guingure roja	199.54 ± 2.52 ^i, A^	129.90 ± 2.65 ^de, C^	65.10	190.71 ± 2.48 ^f, B^	95.57
Morada	257.97 ± 1.27 ^f, A^	130.65 ± 2.34 ^de, C^	50.65	206.64 ± 3.19 ^e, B^	80.10
Conservera	212.77 ± 1.48 ^h, A^	112.54 ± 1.95 ^h, C^	52.89	176.23 ± 3.84 ^g, B^	82.83
Amarilla	299.59 ± 2.84 ^d, A^	142.39 ± 1.67 ^bc, C^	47.53	226.97 ± 3.93 ^d, B^	75.76
Silvestre	293.81 ± 1.93 ^d, A^	136.52 ± 1.75 ^cd, C^	46.47	273.77 ± 2.17 ^b, B^	72.76

Values are expressed as the mean ± standard deviation. Different lowercase letters per column indicate significant statistical difference (*p* ≤ 0.05). Different capital letters per row indicate significant statistical difference (*p* ≤ 0.05). BI: bioaccessibility index.

**Table 6 foods-10-01995-t006:** Total anthocyanin content and bioaccessibility index of 12 ecotypes of *S. purpurea* L. before and after in vitro digestion.

Ecotype	Chemical Extraction (mg C3-GE/100 g)	Gastric Phase (mg C3-GE/100 g)	BI (%)	Intestinal Phase (mg C3-GE/100 g)	BI (%)
Atoyaxi	14.92 ± 1.19 ^a, A^	19.58 ± 0.37 ^a, B^	131.23	ND	0.00
Carnuda	6.35 ± 0.77 ^c, A^	9.24 ± 0.95 ^d, B^	145.51	ND	0.00
Chabacana	ND	ND	0.00	ND	0.00
Chicamerito	7.74 ± 1.54 ^bc, A^	10.56 ± 0.96 ^cd, B^	136.43	ND	0.00
Costeña	7.46 ± 0.15 ^bc, A^	10.65 ± 0.57 ^cd, B^	142.76	ND	0.00
Cuernavaqueña	3.40 ± 0.26 ^d, A^	7.16 ± 0.21 ^e, B^	210.59	ND	0.00
Guingure amarilla	ND	ND	0.00	ND	0.00
Guingure roja	13.14 ± 1.01 ^a, A^	17.58 ± 0.77 ^b, B^	133.79	ND	0.00
Morada	6.35 ± 0.29 ^c, A^	9.17 ± 0.16 ^d, B^	144.41	ND	0.00
Conservera	9.35 ± 0.83 ^b, A^	12.01 ± 0.73 ^cd, B^	128.45	ND	0.00
Amarilla	ND	ND	0.00	ND	0.00
Silvestre	2.00 ± 0.29 ^d, A^	5.61 ± 0.69 ^e, B^	280.50	ND	0.00

ND: Not detected. Values are expressed as the mean ± standard deviation. Different lowercase letters per column indicate significant statistical difference (*p* ≤ 0.05). Different capital letters per row indicate significant statistical difference (*p* ≤ 0.05). BI: bioaccessibility index.

**Table 7 foods-10-01995-t007:** Antioxidant capacity by ABTS of 12 ecotypes of *S. purpurea* L. before and after in vitro digestion.

Ecotype	Chemical Extraction (mM TE/100 g)	Gastric Phase (mM TE/100 g)	Intestinal Phase (mM TE/100 g)
Atoyaxi	6.71 ± 0.07 ^efg, A^	1.19 ± 0.04 ^a, C^	2.97 ± 0.14 ^a, B^
Carnuda	8.23 ± 0.06 ^a, A^	1.01 ± 0.16 ^ab, C^	2.35 ± 0.16 ^abc, B^
Chabacana	7.28 ± 0.18 ^cd, A^	0.88 ± 0.13 ^bc, C^	1.93 ± 0.03 ^bc, B^
Chicamerito	7.65 ± 0.16 ^bc, A^	0.68 ± 0.04 ^cd, C^	2.43 ± 0.12 ^abc, B^
Costeña	6.35 ± 0.11 ^g, A^	1.01 ± 0.05 ^ab, C^	2.18 ± 0.06 ^abc, B^
Cuernavaqueña	6.80 ± 0.14 ^ef, A^	0.97 ± 0.05 ^ab, C^	2.55 ± 0.21 ^abc, B^
Guingure amarilla	5.18 ± 0.13 ^h, A^	0.52 ± 0.12 ^d, C^	1.82 ± 0.21 ^c, B^
Guingure roja	6.51 ± 0.21 ^efg, A^	1.04 ± 0.09 ^ab, C^	2.87 ± 0.14 ^ab, B^
Morada	7.71 ± 0.19 ^b, A^	1.25 ± 0.16 ^a, C^	2.72 ± 0.08 ^abc, B^
Conservera	7.39 ± 0.11 ^bc, A^	0.88 ± 0.02 ^bc, C^	2.29 ± 0.06 ^abc, B^
Amarilla	6.88 ± 0.05 ^de, A^	0.83 ± 0.05 ^bc, C^	2.82 ± 0.17 ^abc, B^
Silvestre	6.43 ± 0.17 ^fg, A^	0.52 ± 0.10 ^d, C^	2.09 ± 1.11 ^abc, B^

Values are expressed as the mean ± standard deviation. Different lowercase letters per column indicate a significant statistical difference (*p* ≤ 0.05). Different capital letters per row indicate a significant statistical difference (*p* ≤ 0.05).

**Table 8 foods-10-01995-t008:** Antioxidant capacity by DPPH of 12 ecotypes of *S. purpurea* L. before and after in vitro digestion.

Ecotype	Chemical Extraction (mM TE/100 g)	Gastric Phase (mM TE/100 g)	Intestinal Phase (mM TE/100 g)
Atoyaxi	1.79 ± 0.04 ^bcd, A^	1.05 ± 0.05 ^ab, C^	1.49 ± 0.13 ^a, B^
Carnuda	1.89 ± 0.02 ^a, A^	0.99 ± 0.22 ^ab, C^	1.48 ± 0.15 ^a, B^
Chabacana	1.84 ± 0.03 ^abc, A^	0.95 ± 0.28 ^b, C^	1.36 ± 0.15 ^a, B^
Chicamerito	1.87 ± 0.01 ^ab, A^	0.93 ± 0.15 ^b, C^	1.45 ± 0.14 ^a, B^
Costeña	1.86 ± 0.04 ^abc, A^	1.26 ± 0.13 ^ab, C^	1.57 ± 0.08 ^a, B^
Cuernavaqueña	1.88 ± 0.01 ^a, A^	1.14 ± 0.25 ^ab, C^	1.43 ± 0.16 ^a, B^
Guingure amarilla	1.86 ± 0.01 ^abc, A^	1.42 ± 0.05 ^a, C^	1.51 ± 0.14 ^a, B^
Guingure roja	1.71 ± 0.06 ^d, A^	1.18 ± 0.01 ^ab, C^	1.37 ± 0.26 ^a, B^
Morada	1.86 ± 0.02 ^abc, A^	1.18 ± 0.13 ^ab, C^	1.46 ± 0.12 ^a, B^
Conservera	1.86 ± 0.01 ^abc, A^	1.29 ± 0.09 ^ab, C^	1.55 ± 0.16 ^a, B^
Amarilla	1.87 ± 0.01 ^ab, A^	1.23 ± 0.14 ^ab, C^	1.56 ± 0.07 ^a, B^
Silvestre	1.78 ± 0.03 ^cd, A^	0.99 ± 0.07 ^ab, C^	1.28 ± 0.02 ^a, B^

Values are expressed as the mean ± standard deviation. Different lowercase letters per column indicate a significant statistical difference (*p* ≤ 0.05). Different capital letters per row indicate a significant statistical difference (*p* ≤ 0.05).

**Table 9 foods-10-01995-t009:** Pearson correlation coefficients among the antioxidant capacity and antioxidant compounds of 12 ecotypes of *S. purpurea* L. before and after in vitro digestion.

	Before In Vitro Digestion		After In Vitro Digestion			
			Gastric Phase		Intestinal Phase	
	ABTS	DPPH	ABTS	DPPH	ABTS	DPPH
TCC	0.44	0.62	−0.48	−0.58	−0.30	0.22
AA	0.65	0.43	0.83	0.84	0.44	0.29
TPC	0.94	0.83	−0.27	−0.30	0.81	0.86
TFC	0.91	0.82	−0.18	−0.28	0.82	0.83
TAC	0.82	0.73	−0.15	−0.41	−0.25	−0.61

TCC: Total carotenoid content, AA: Total ascorbic acid content, TPC: total phenolic content, TFC: Total flavonoid content, TAC: total anthocyanin content, ABTS: antioxidant capacity with ABTS assay, DPPH: antioxidant capacity with DPPH assay.

## Data Availability

Not applicable.

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
