# Peer review of "Effect of In Vitro Digestion on the Antioxidant Compounds and Antioxidant Capacity of 12 Plum (*Spondias purpurea* L.) Ecotypes"

_foods, 2021, doi:10.3390/foods10091995_

Round 1
Reviewer 1 Report
The authors investigated the effect of in vitro gastrointestinal digestion on the contents of antioxidant compounds and the antioxidant capacity of 12 plum Spondias purpurea L. ecotypes. Two stages of digestion (gastric and intestinal stages) were identified and the bioaccessibility index (BI) was obtained. The overall study design is clear, and the aims have been achieved, however, several major issues need to be addressed.
Comments:
- Title: The study title seems a little awkward. The authors may consider changing “the carotenoids, ascorbic acid, phenolic compounds” to a general term “antioxidant compounds”.
- Abstract: The last sentence “TPC and TFC presented positive and high correlation with ABTS (r ≥ 0.8) and DPPH (r ≥ 0.8)” is not clear. The labeled correlation represents for which one of the antioxidant compounds?
- In the introduction section the authors could add more background information regarding how different stages of digestion may influence the antioxidant compounds and capacity of the S. Purpurea fruits. In addition, the authors can introduce ABTS and DPPH with more details and add their full terms.
- The language of the manuscript needs to be improved a lot. Many long sentences are present especially in the introduction, i.e., Line 46-51, sentences in the parentheses could be listed individually.
- Results 3.1: Is there any evidence from previous literature regarding the statements for pectin (line 225-230)? Line 226: “studies have shown…” the authors need to give the references.
- Line 227-230: This sentence needs rework. It reads like pectin could enhance micelle formation.
- Since pectin is a soluble dietary fiber, it cannot be digested in the small intestine, so it should exert the same effect on lipid binding and on the incorporation of pectin. In this case, why would the results for these two stages different for TCC? Why it was not detected in the gastric phase while detected again in the intestinal phase?
- Line 240: Explain more regarding how these factors may influence the digestive process.
- For all the reported results, did the authors conduct statistical analyses for the among-group differences? It seems that only the within group (before and after digestion) results were exerted for statistical analysis. However, if the authors would like to compare the results among these 12 ecotypes, it’s better to also performed analysis for these data. Further, why wasn’t statistical analysis performed for BI?
- Line 266: “enable” bioaccessibility does not make sense. Did the authors intend to say “enhance?”
- Line 272-175: This paragraph repeats the content of line 260. Move above with more explanations or remove.
- Line 306-308: Again, repeated contents. Combine this to line 294 with more explanations.
- Line 352: Did the authors mean the range was 37.7-77.57%? Similar mistake also found in line 393.
- Line 362-364: Why would this be a possible cause? How such reaction influences the BI in the intestinal phase?
- Line 428: Why was the BI of anthocyanins after gastric digestion elevated? This result was different from all others. The authors could give more explanations regarding such observation.
Author Response
- Title: The study title seems a little awkward. The authors may consider changing “the carotenoids, ascorbic acid, phenolic compounds” to a general term “antioxidant compounds”.
Response: Title was changed. Changes made to the manuscript are highlighted in red (lines 2-3).
- Abstract: The last sentence “TPC and TFC presented positive and high correlation with ABTS (r ≥ 0.8) and DPPH (r ≥ 0.8)” is not clear. The labeled correlation represents for which one of the antioxidant compounds?
Response: The sentence indicate that in the intestinal stage, total phenolic content (TPC) and Total flavonoid content (TFC) presented positive and high correlation with ABTS (r ≥ 0.8) and DPPH (r ≥ 0.8) respectively. See Table 9.
- In the introduction section the authors could add more background information regarding how different stages of digestion may influence the antioxidant compounds and capacity of the purpurea fruits. In addition, the authors can introduce ABTS and DPPH with more details and add their full terms.
Response: We consider that this is included in results and discusión.
- The language of the manuscript needs to be improved a lot. Many long sentences are present especially in the introduction, i.e., Line 46-51, sentences in the parentheses could be listed individually.
Response: The English writing was revised and improved. A certificate is attached to guarantee it.
- Results 3.1: Is there any evidence from previous literature regarding the statements for pectin (line 225-230)? Line 226: “studies have shown…” the authors need to give the references.
Response: References were included [27-28]. Changes made to the manuscript are highlighted in red (line 228-233). References:
- Cervantes, P.B.; Ornelas, P.J.; Pérez, M.J.; Ruiz, C.S.; Rios, V.C.; Ibarra, J.V.; Yahia, M.E.; Gardea, B.A. Effect to pectin on lipid digestion and possible implications for carotenoid bioavailability during pre-absortive stanges: A review. Food Res Int 2017, 99, 917-927.
- Cervantes, P.B.; Ornelas, P.J.; Pérez, M.J.; Reyes, H.J.; Zamudio, F.P.; Rios, V.C.; Ibarra, J.V.; Ruiz, C.S. Effect to pectin concentration and properties on digestive events involved on micellarization of free and esterified carotenoids. Food Hydrocoll 2016, 60, 580-588.
- Line 227-230: This sentence needs rework. It reads like pectin could enhance micelle formation.
Response: Changes made to the manuscript are highlighted in red (lines 228-233).
- Since pectin is a soluble dietary fiber, it cannot be digested in the small intestine, so it should exert the same effect on lipid binding and on the incorporation of pectin. In this case, why would the results for these two stages different for TCC? Why it was not detected in the gastric phase while detected again in the intestinal phase?
Response: The results for these two stages were different for TCC, because not only does the pectin content influence, but also the change of pH and enzymes. This is explained in lines 253-261.
- Line 240: Explain more regarding how these factors may influence the digestive process.
Response: Changes made to the manuscript are highlighted in red (lines 243-246).
- For all the reported results, did the authors conduct statistical analyses for the among-group differences? It seems that only the within group (before and after digestion) results were exerted for statistical analysis. However, if the authors would like to compare the results among these 12 ecotypes, it’s better to also performed analysis for these data. Further, why wasn’t statistical analysis performed for BI?
Response: The statistical analysis was included to compare the results among the 12 ecotypes. See Tables 2-8. The statistical analysis for BI is not included because this analysis is not made in the literature. See:
- Correa-Betanzo J, Allen-Vercoe E, McDonald J, Schroeter K, Corredig M, Paliyath G. Stability and biological activity of wild blueberry (Vaccinium angustifolium) polyphenols during simulated in vitro gastrointestinal digestion. Food Chem. 2014. 165:522-531.
- Flores, Floirendo & Singh, Rakesh & Kerr, William & Pegg, Ronald & Kong, Fanbin. (2014). Total phenolics content and antioxidant capacities of microencapsulated blueberry anthocyanins during in vitro digestion. Food Chemistry. 153. 272–278.
- Schulz M, Biluca FC, Gonzaga LV, Borges GD, Vitali L, Micke GA, de Gois JS, de Almeida TS, Borges DL, Miller PR, Costa AC, Fett R. Bioaccessibility of bioactive compounds and antioxidant potential of juçara fruits (Euterpe edulis Martius) subjected to in vitro gastrointestinal digestion. Food Chem. 2017. 228: 447-454.
- Line 266: “enable” bioaccessibility does not make sense. Did the authors intend to say “enhance?”
Response: Changes made to the manuscript are highlighted in red (line 272).
- Line 272-275: This paragraph repeats the content of line 260. Move above with more explanations or remove.
Response: This paragraph was eliminated.
- Line 306-308: Again, repeated contents. Combine this to line 294 with more explanations.
Response: This paragraph was eliminated and more explanation was added (lines 301-307).
- Line 352: Did the authors mean the range was 37.7-77.57%? Similar mistake also found in line 393.
Response: Changes made to the manuscript are highlighted in red (lines 353 and 396-397).
- Line 362-364: Why would this be a possible cause? How such reaction influences the BI in the intestinal phase?
Response: Changes made to the manuscript are highlighted in red (lines 365-366).
- Line 428: Why was the BI of anthocyanins after gastric digestion elevated? This result was different from all others. The authors could give more explanations regarding such observation.
Response: Changes made to the manuscript are highlighted in red (line 417-426).

Reviewer 2 Report
The article entitled “Effect of in vitro digestion on the carotenoids, ascorbic acid, phenolic compounds and antioxidant capacity of 12 plum (Spondias purpurea) ecotypes” investigated the effect of in vitro gastrointestinal digestion on the carotenoids, ascorbic acid, total phenolics compounds, total flavonoids, anthocyanins, and antioxidant capacity of twelve plum Spondias purpurea L. ecotypes.
The article is written well. Results are well discussed, and the results support conclusions. However, the major weakness of this investigation is the spectrophotometric methods used for the analysis of carotenoids, ascorbic acids, and anthocyanins. High-Performance Liquid Chromatography (HPLC) method is generally used to obtain the result with better accuracy.
I have the following other comments for this article:
In the title, “Spondias purpurea” can be replaced with “Spondias purpurea L.”
The keywords Spondias purpurea, ecotypes, and in vitro digestion are already mentioned in the title. Please suggest the keywords which are not mentioned in the title.
Author Response
In the title, “Spondias purpurea” can be replaced with “Spondias purpurea L.”
Response: Spondias purpurea” was changed with “Spondias purpurea L. Changes made to the manuscript are highlighted in red (line 3).
The keywords Spondias purpurea, ecotypes, and in vitro digestion are already mentioned in the title. Please suggest the keywords which are not mentioned in the title.
Response: The keywords were changed with “phenolic compounds, carotenoids, ascorbic acid, ABTS, DPPH”. Changes made to the manuscript are highlighted in red (line 25).

Round 2
Reviewer 1 Report
All changes was made.